



# Characterizing the 2015 Indonesia Fire Event Using Modified MODIS Aerosol Retrievals

Yingxi R. Shi[1,3], Robert C. Levy[1], Thomas F. Eck[1,3], Brad Fisher[1,2], Shana Mattoo[1,2], Lorraine A. Remer[4], Ilya Slutsker[1,2],

and Jianglong Zhang[5]

[1]NASA Goddard Space Flight Center, Greenbelt, MD, USA
[2]SSAI, Lanham, MD, USA
[3]GESTAR, USRA, Columbia, MD, USA
[4]UMBC/JCET, Baltimore, MD, USA
[5]UND, Grand Forks, ND, USA

*Correspondence to*: Yingxi Shi (yingxi.shi@nasa.gov)

**Abstract.** The Indonesian fire and smoke event of 2015 was an extreme episode that affected public health and caused severe economic and environmental damage. The MODIS Dark Target (DT) aerosol algorithm, developed for global applications, significantly underestimated regional aerosol optical depth (AOD) during this episode. The larger-than-global-averaged uncertainties in DT product over this event were due to both an overly zealous set of masks that mistook heavy smoke plumes for clouds and/or inland water, and also an aerosol model developed for generic global aerosol conditions. Using Aerosol Robotic Network (AERONET) Version 3 sky inversions of local AERONET stations, we created a specific aerosol model for the extreme event. Thus, using this new less-absorbing aerosol model, cloud masking based on results of the MODIS cloud optical properties algorithm, and relaxed thresholds on both inland water tests and upper limits of the AOD retrieval we created a research algorithm, and applied it to 80 appropriate MODIS granules during the event. Collocating and comparing with AERONET AOD shows that the research algorithm doubles the number of MODIS retrievals greater than 1.0, while also significantly improving agreement with AERONET. The final results show that the operational DT algorithm had missed approximately 0.22 of the regional mean AOD, but as much as AOD = 3.0 for individual 0.5° grid boxes. This amount of missing AOD can skew the perception of the severity of the event, affect estimates of regional aerosol forcing, and alter aerosol modeling and forecasting that assimilate MODIS aerosol data products. These results will influence the future development of the global DT aerosol algorithm.

## 1 Introduction

Extreme aerosol events, as a result of severe biomass burning, have large regional and global impacts. The biomass burning causes destruction in ecosystems, disruption to economics, and harms public health. For example, the El Nino-related 2015 Indonesia fire (Field et al., 2016) event released 1750 million metric tons of carbon dioxide, which is equal to 5.5% of the global carbon emission from fossil fuel and industrial processes in 2010 (Parker et al., 2016; Glauber et al. 2016; IPCC, 2014). The five months of burning also caused significant economic and environmental damage, including $16.1 billion in




economic losses (Glauber et al. 2016), 2.6 million hectares of Indonesian land burned, as well as destruction of fragile peatland ecosystems (Lohberger et al., 2017). Studies also show that public health was harmed via accumulated and/or transported smoke (Marlier et al., 2015; Crippa et al., 2015). The long-term effects of the smoke are estimated to have caused an additional 100,000 mortalities across Indonesia, Malaysia, and Singapore (Koplitz et al., 2016).

5  Due to the vast destruction and long-lasting impacts, the research, applied science and policy communities have attempted to observe, understand, simulate, and predict events like the Indonesian fires. Satellite aerosol products are one important data source used by a wide range of disciplines in general studies of fire and smoke. Examples of applications of satellite products include: fire intensity estimation (Petrenko et al., 2012), aerosol transport modeling and visibility forecasts (Collins et al., 2001; Zhang et al., 2008), air quality prediction (Al-Saadi et al., 2005; Wang et al., 2003), and human health
10  assessments (Van Donkelaar et al., 2010; Lighty et al., 2000). Satellite aerosol products, especially those derived from the passive sensors, have difficulties retrieving aerosol signals when the smoke plumes are very optically thick (van Donkelaar et al., 2011; Zhang et al., 2016; Witte et al., 2011). Very optically thick aerosol plumes, defined as having aerosol optical depth (AOD, symbol $\tau$) greater than 3.0, which may have high visible reflectance and high spatial variability near the source region, could be misclassified as clouds or other features. By excluding these optically thick aerosol data, this
15  misclassification can introduce a low bias in aerosol regional climatology and further influence other studies that rely on satellite data. In particular the aerosol modeling and aerosol data assimilation efforts to model and predict the consequences of these events for air quality and visibility forecasts will be misled due to this low bias in the "observed" quantities (Zhang et al., 2006; Benedetti, et al., 2009; Chung et al., 2010). This was indeed the case during the Indonesian smoke event of 2015.

20  In this study, we focused on a domain and temporal period defined as -10° to 10° N and 95° to 125° E from August to October 2015 when intense burning existed. We noted that the operational MODerate Resolution Imaging Spectroradiometer (MODIS) aerosol products had trouble capturing the complete picture of aerosol loading during the 2015 Indonesia burning event. Therefore, we developed a research algorithm to address this problem and to bring back those missing retrievals. This research algorithm is based on the operational Dark Target (DT) aerosol algorithm but modified
25  with a new cloud mask and a new aerosol model generated from local AERONET inversion products. We applied the new research algorithm and evaluate results against AERONET version 3 AOD. This new regional aerosol climatology was investigated using the research product and statistical analyses were also conducted to understand the aerosol distribution over this event.



## 2 Remote sensing of aerosol and clouds over Indonesia

### 2.1 MODIS Dark Target aerosol algorithm

The MODIS Dark Target algorithm for retrieving aerosol properties over land utilizes three wavelengths (0.47, 0.66, and 2.1 μm) over dark vegetation covered land surfaces following the look up table (LUT) method (Levy et al., 2007ab, 2013). The

5   algorithm applies two fundamental assumptions that allow constraint of surface reflectance and aerosol properties (aerosol model) in order to retrieve the AOD. The first assumption concerns estimating the surface reflectance referring to an assumed relationship between the surface reflectance at 2.1 μm and the surface reflectance in the visible (Levy et al., 2007b). The second assumption concerns predetermining a fine mode aerosol model prescribed for every season and region. Prior knowledge of aerosol model is determined via global analyses of AERONET sky radiance inversion products before

10   February 2005 (Levy et al., 2007a). Based on the reported dominant aerosol type at AERONET sites at that time, regions seasonally dominated by "strongly absorbing" or "non-absorbing" aerosol types are identified and set aside, while everywhere else is assigned the "moderately absorbing" aerosol model. There were no AERONET sites established in or near Indonesia before 2005. Thus, the preselected aerosol model for Indonesia in the operational DT algorithm is the moderately absorbing aerosol model with a single scattering albedo of 0.92 (Levy et al., 2007a).

15   The MODIS DT algorithm first groups the input radiances into arrays of 20x20 pixels at 500 m resolution, which is nominally a 10 km box at nadir. Within this retrieval box the algorithm proceeds with a cascade of screening procedures to remove pixels that will violate the fundamental assumptions about surface properties and aerosol model. These screening procedures include a cloud mask, a snow and ice filter, an inland water test, and elimination of bright surfaces. Two masks that are particularly relevant for our situation of thick smoke over Indonesia are cloud mask and inland water test. The DT

20   cloud screening procedure relies on tests that compare the absolute value and the spatial variability of top-of-atmosphere (TOA) reflectance at 0.47 μm at 500-meter resolution with a threshold value. Pixels that are "too bright" or "too variable" are masked as clouds. In addition, the algorithm makes use of the 1.38 $\mu$m TOA reflectance at 1 km resolution to identify and mask cirrus.

The inland water mask is basically the Normalized Difference Vegetation Index (NDVI), defined as Eq. (1):

25   $$\text{NDVI} = {\rho_{0.87} - \rho_{0.66}}\big/{\rho_{0.87} + \rho_{0.66}} \tag{1}.$$

where ρ is the reflectance at TOA at 0.87 and 0.66 μm, as subscripted. In addition to separating vegetated and non-vegetated surfaces, NDVI is sensitive to a thin layer of water on the surface, such as snow melting or swamp surfaces. NDVI can also be used to remove pixels near cloud edges. The operational DT algorithm requires NDVI > 0.1 for the retrieval to be performed, which enables the identifications of "ideal" dark-land targets for DT retrieval and avoids situations that would

30   introduce large uncertainties in the retrieval.

After the screening process has removed clouds and various other surfaces in violation of the algorithm's assumptions, the retrieval returns to the remaining "good" pixels in the 10 km retrieval box and discards the brightest 50% and the darkest





20% of these qualified pixels, defined using the reflectance at 0.66 μm. If there are at least 12 pixels remaining after this vigorous selection process, these remaining pixels are aggregated to produce the average TOA spectral reflectance representative of the 10 km box. From this aggregation, the inversion is performed based on the pre-calculated LUT. The pre-calculated LUT only extends to AOD of 5.0 (at 0.55 μm). For an algorithm that aims to retrieve aerosol globally, this

AOD cap is reasonable (Remer et al., 2008). The retrieved DT AOD over land has an expected uncertainty of ± (0.05+15%AOD).

While these assumptions and screening procedures are appropriate for a global, operational DT algorithm, we have found that there are exceptional situations. The Indonesian smoke event is one of these exceptional situations that require modification of the operational DT algorithm to obtain accurate retrievals (or even to retrieve in the first place). Without

modifying the global thresholds for masking, the missing retrievals will lead to a product with a statistically low bias. In developing these modifications, we will require additional information from other sensors and algorithms to identify heavy smoke plumes, help separate aerosol from cloudy scenes and provide information about aerosol model, as well as provide validation for any improvements we implement.

### 2.2 MODIS Deep Blue aerosol algorithm

There is a second MODIS aerosol retrieval algorithm known as Deep Blue (DB). The MODIS DB algorithm was first designed to retrieve aerosol over arid and semi-arid regions and later was later extended to vegetated surfaces (Hsu et al., 2006, Hsu et al., 2013). The DB method, in part, relies on aerosol light absorption for such aerosol types as dust and smoke at 0.412 and 0.47 μm wavelengths. Instead of aggregating the TOA reflectance to 10 km resolution first, the DB algorithm retrieves AOD at 1 km resolution then aggregates AOD to 10 km. The DB algorithm uses a pre-existing database of surface

properties based on location, season, scattering angle, and the greenness of the ground (Hsu et al., 2013). The reported uncertainties of the highest quality DB retrievals (QA = 3) is defined as Eq. (2):

$$\pm([0.086 + 0.56\tau_{DB}]/[1/\mu_0 + 1/\mu])  \tag{2}$$

where $\mu_0$ and $\mu$ are the cosine of the solar and view zenith angles, respectively (Sayer et al., 2013).

### 2.3 OMI UV aerosol index

Although optically thick smoke looks very similar to clouds throughout most of the visible spectrum, smoke and clouds appear very different at both shorter (ultraviolet, UV) and longer (near-infrared, IR) wavelengths. Due to their strong absorption in the UV and near UV wavelengths, smoke particles can be easily detected using observations in the UV spectrum, such as the Ozone Monitoring Instrument (OMI) UV aerosol index (AI).

The Ozone Monitoring Instrument is installed on the Aura satellite, which is part of the A-train constellation that follows

Aqua (crossing equator at approximately 13:30 local solar time). OMI spans a broad swath of 2600 km with a hyperspectral coverage from UV to visible (0.264 to 0.504 μm) with a spatial resolution of 13 × 24 km at nadir (Levelt et al., 2006). In



this study, we use the UV aerosol index (AI), which is reported within the OMI OMAERUV product (Torres et al., 2007; 2013). Near-zero AI values indicate clouds. Positive AI values represent the presence of UV-absorbing aerosols that can be black carbon, mineral dust, or volcanic ash. Some non UV-absorbing small aerosol particles such as sulfate aerosols, can also result in small negative AI, but the signal is much weaker. Because OMI and MODIS have different spatial resolutions

and fields of view, we use the OMI-MODIS collocation aerosol product (OMMYDAGEO), developed by the OMI science team. OMMYDAGEO product provides the OMI along track and cross track indices for every overlapping pixel in the MODIS granule at both 3 and 10 km resolution for the two Level2 MODIS aerosol products, MYD04_L2 and MYD04_3km (J. Joiner, 2017).

In this study, the OMI AI is used to identify heavy biomass burning/smoke plumes at coarse resolution. Strong positive AI

values over Indonesia indicate the potential presence of heavy smoke aerosols, although high AI values may also indicate aerosol above or aerosol mixed with cloud cases. Thus, OMI AI values are only used as the first step for identifying heavy smoke plumes that can further be "rescued". Note that OMI instrument suffers a row anomaly issue after 2008, which results in data gaps within a granule (OMIRA Team, 2012). The impact of missing AI data introduced by the row anomaly on this study is discussed in detail in the end of Section 4.0.

**2.4 MODIS cloud optical properties algorithm**

Smoke particles are much smaller in size than cloud particles, so smoke particles appear nearly transparent at longer wavelengths. Thus, in tandem with using shorter UV wavelengths (OMI AI) to identify potential smoke plumes, observations from IR and near-IR channels can be used to exclude aerosol above or mixed with cloud scenes. The MODIS cloud optical properties algorithm uses visible, near-IR, and thermal IR channels to retrieve cloud physical and radiative

properties at 1 km resolution (Platnick et al., 2003). Examples of the retrieved parameters are cloud thermodynamic phase, cloud particle effective radius, and cloud optical thickness along with retrieval quality flags. Like the DT algorithm, the cloud algorithm makes assumptions about the scene it is retrieving. When those assumptions are violated the retrieval fails and returns an error flag. The first assumption to be tested is that the scene must contain a valid cloud top pressure, which is derived using thermal IR channels. Then there are three retrieval failure metrics in the MODIS cloud product indicating that

the retrieval of cloud droplet or crystal effective radius failed at one of these three wavelengths, 1.6, 2.1 and 3.7 μm, respectively. If there is no cloud in the scene and only smoke, there may be no valid cloud top pressure retrieved, and the cloud optical properties algorithm will not produce a retrieval. Even if a retrieval is attempted on the smoke, smoke particle sizes are orders of magnitude smaller than cloud droplet or crystal sizes. The cloud effective radius retrievals will be out of bounds of the assumptions, the retrieval will fail and the failure metrics will be set. Thus, these metrics and the lack of a

cloud retrieval can be used to separate smoke from clouds (personal communication with Dr. Gala Wind).



**2.5 AERONET sun/sky aerosol products**

The AErosol RObotic NETwork (AERONET) is a global aerosol-monitoring network of sun/sky observing radiometers that is commonly used as a benchmark for validating satellite-retrieved AOD (Holben et al., 1998; Levy et al., 2010, 2013; Remer et al., 2005; Sayer et al., 2013; Zhang and Reid, 2006, Shi et al., 2011). The instruments measure attenuated solar energy through two modes: direct sun and scanning sky (Holben et al., 1998). The direct sun measurement mode provides an observation of spectral AOD every 3 or 15 minutes (depending upon instrument version and settings). Multiple quality assurance steps as well as vigorous cloud screening procedures are applied to the version 2 Level 2.0 AOD data to ensure a high-quality data set. However, the cloud screening removes many high AOD observations and introduces a low bias to the version 2 data set (Eck et al., 2016). Thus, to validate our research algorithm described in this paper we turn to the version 3 AERONET level 2 data that specifically include these high AOD cases (Eck et al., 2016; Eck et al., 2018). This is particularly important for the heavy smoke during the Indonesian event studied here. The version 3 AERONET data also tend to have less thin cirrus contamination and better quality-control algorithms than does version 2. The AOD uncertainty in version 3 Level 2 data is practically the same as in Version 2, which is ~0.01 in the visible and near-infrared wavelengths and increasing to ~0.02 in the UV (Eck et al., 1999). Cloud screening in Version 3 is briefly described in Eck et al. (2018) and in depth in a future paper (D. Giles personal communication). In this study, we will use data from the following AERONET sites: Jambi, Palangkaraya, Kuching, Pontianak, and Singapore. Figure 1 shows the geolocation of these five sites.

Besides using AERONET AOD products from the direct sun measurements for validation of satellite AOD, we also make use of the AERONET inversion products from the sky scanning measurements. Aerosol inversion products include aerosol microphysical properties such as particle size distribution, complex refractive index, and phase function (Dubovik and King, 2000; Dubovik et al., 2002; 2006). We use inversions from the almucantar mode sky measurements to build a regional smoke aerosol model. The almucantar mode is a series of measurements of the sky, spanning all azimuthal angles ($0^\circ$ to $\pm$ $180^\circ$), at a fixed zenith angle equal to solar zenith angle (SZA). This creates a set of measurements across a range of scattering angles (Holben et al., 1998). The limitation of the almucantar mode is that when SZA is smaller than 50°, the range of scattering angles is small, leading to potentially large measurement error (Holben et al., 2006). We used inversion products following QA procedures in Holben et al., (2006), except that we use more strict threshold of AOD at 0.675 μm ($AOD_{0.675}$) greater than 0.4 rather than AOD at 0.44 μm as AERONET team recommended.

**3 Case study: An intense high AOD smoke event on September 22nd 2015**

To illustrate how missing retrievals can create a low bias in regional MODIS AOD estimates, we focus on a fire event that took place near Kalimantan on the island of Borneo on September 22nd, 2015. Figure 2a shows the MODIS RGB image cropped to -5° to 5° latitude and 105° to 120° longitude. Significant smoke aerosol plumes, in yellowish grey colour, can be observed across the image and are clearly distinguishable from the white clouds observed surrounding the smoke plume.





The Palanakaraya AERONET site (marked with a blue star) within this scene is under extremely high smoke concentrations on this day. In fact, the AOD is so high at this site and date that there is no signal at 500 nm and even at 675 nm for most of the day (nearly complete attenuation). The AOD at 875 nm averaged 4.3 over the nearly 2 hours of available measurements, and the average Angstrom Exponent from 870-1640 nm over this same interval was 1.85, thus indicating fine mode smoke

particles and not cloud contamination. Figure 2b and d show the corresponding aerosol retrievals from the DT and DB aerosol products and Fig. 2c shows the OMI AI values. Note that all retrievals from DT and DB aerosol products are used here without further quality assurance filtering. Over heavy aerosol regions that have OMI AI values exceeding 3.0, aerosol retrievals are mostly missing from the MODIS DT aerosol products and are partially missing from the MODIS DB aerosol products. Figure 2 demonstrates that passive sensor observations in visible wavelengths may have trouble separating heavy

aerosol plumes from clouds. In comparison, the OMI AI can be used effectively to qualitatively detect thick UV-absorbing aerosol plumes that are missed by MODIS DT and DB aerosol products. However, only using OMI AI cannot identify smoke above clouds and thus, further analyses are performed to separate aerosols from clouds.

The MODIS DT algorithm fails to retrieve AOD over the thickest part of the plume, is because the NDVI mask and the internal cloud mask have filtered out the optically thick smoke pixels. Within the region where heavy smoke plumes exist,

the NDVI value ranges from 0.0 to 0.1 (Fig. 2e), which is below the operational threshold of 0.1. As we mentioned before, the threshold of NDVI > 0.1 is set to ensure an optimum retrieval condition, which will require adjustment to allow retrievals over optically thick smoke. The operational internal cloud mask also screens out the smoke plume. Thus, a "call back" method is needed to distinguish aerosols from clouds in regions where thick plumes exist. As described in Section 2.4, based on the differences between particle sizes of aerosols and clouds, the MODIS cloud optical property retrievals typically

fail when applied to optically thick smoke regions, and these failures are recorded in failure metrics. Note that the failure metrics are only available when a scene is a priori identified as a cloud. Thus, these failure metrics will only help detect misidentified smoke plumes when the aerosol is sufficiently thick to resemble a cloud by other tests. We examine these failure metrics from the MODIS cloud product at 1 km resolution. If no successful cloud retrieval is reported for attempts made using any of the three possible wavelengths, we consider the pixel to be an aerosol polluted, cloud free pixel.

Examples of those misidentified smoke pixels are shown in red in Fig. 2f. Plotted on top of the true color image, red pixels are pixels with failed cloud retrievals and are only visible above optically thick aerosol plumes (Fig. 2f). Thus, these metrics will be used in combination with the aerosol algorithm's operational cloud mask to identify cloud-free scenes with low to moderate aerosol loading and to reclassify scenes as cloud-free in high aerosol loading when the operational mask initially designates the scene as cloudy.

This case study demonstrates that the standard MODIS DT aerosol algorithm is missing a large fraction of the heavy smoke from Indonesian fires in 2015, partially due to very low NDVI values over the thick smoke regions and partially due to a very stringent cloud screening algorithm. This case study also suggests that OMI AI is able to identify the heavy smoke unencumbered by these constraints and that the MODIS cloud optical properties product can be used for distinguishing between heavy smoke and clouds.



#### 4 An aerosol algorithm for heavy smoke

Based on the case study, we have investigated a method for "rescuing" heavy smoke pixels for the operational MODIS DT products. This process is initiated by constructing a NDVI mask. Note a NDVI threshold of 0.1 is used in the operational MODIS DT algorithm. For regions with NDVI values in between -0.02 to 0.1, observed areas could include one of the

5 following scenarios such as coastal areas, surface with standing water, arid/desert surfaces, urban surfaces with haze, aerosols near cloud edges, and very optically thick aerosol plumes. The study region does not contain a large fraction of deserts or highly urban surfaces. Thus, regions with NDVI values less than 0.1 are likely to be regions such as coastal areas, aerosols near cloud edges, and very optically thick aerosol plumes. Furthermore, sensitivity studies (not shown) suggest that an NDVI threshold of 0.01 can be effectively used to remove coastal regions while maintaining most of the optically thick

aerosol plumes (e.g. See Fig. 2e). Thus, pixels with NDVI values of 0.01-0.1 are considered as potential thick smoke aerosol pixels and are selected for further study.

Correspondingly, a modified cloud mask is also implemented. Here, a pixel is identified as suitable for applying aerosol retrieval algorithm if one or both of the following criteria are met: (a) the pixel passed the cloud screening steps based on the aerosol DT algorithm or (b) the pixel is both identified as a "cloud pixel" by the operational aerosol DT algorithm and also

failed to produce a cloud optical property retrieval based on the cloud failure metrics. In this way, some pixels that were previously removed due to cloud screening steps are reconsidered for aerosol retrievals. The modified cloud screening method as described above can still identify cloud pixels outside the heavy smoke regions. Within the heavy smoke regions, smoke pixels that were previously misidentified as "cloud pixels" can be successfully labeled as smoke pixels, with the use of the modified cloud screening method.

In addition, AOD at 0.55 μm ($AOD_{055}$) of 5 is currently used as the upper limit for the operational MODIS DT retrievals. Retrievals that require extrapolation beyond AOD = 5, return a fill value and are not retrieved. Thick smoke plumes for the study period can have AOD values exceeding this threshold. Thus, we explored removing this upper limit. With this change, the algorithm is allowed to extrapolate the LUT to retrieve higher AOD values. However, due to the limited sensitivity of the MODIS sensor under very thick smoke plume conditions, we found that the retrieval had little skill at distinguishing

between different AODs greater than 5. Therefore, we continue to constrain AOD to 5 in our validation, which means all retrieved AOD greater than 5 are assigned to 5 during the validation. We understand that this requirement could introduce underestimation of AOD because AERONET has reported AODs greater than five during this event, however, we took this precautious step due to the limitation in MODIS sensor.

Besides the above steps to enable aerosol retrievals over thick smoke plumes, an additional step is also implemented to

30 improve retrieval accuracy. A localized aerosol model is needed for retrievals with very high AOD values as small changes in aerosol properties can introduce large errors in AOD retrievals (Ichoku et al., 2003). Thus, we will re-examine the aerosol model used by the operational algorithm for the region of interest for the given season. The "moderately absorbing fine





mode aerosol model" (Levy et al., 2007a) is used for this region in the current operational MODIS DT algorithm. This is a generic model derived from data in other parts of the world and never specifically evaluated for smoke aerosols in Indonesia. When the operational DT aerosol models were first developed (Levy et al., 2007a), there were insufficient AERONET sites available for deriving a region-specific aerosol model for Indonesia. Now there are AERONET stations in Indonesia that are

active during the smoke season. In this study, a localized smoke aerosol model is developed by using AERONET (version 3, Level 2) derived size distribution and the refractive index for the study period of August to October 2015 for the five stations identified in Fig. 1. The size distribution and the refractive index are analyzed as functions of AOD at 0.675 μm ($AOD_{0.675}$). Figure 3 shows the volume size distributions of 163 inversions divided into 22 particle radii sorted as a function $AOD_{0.675}$ into bins of 0-0.2, 0.2-0.4, 0.4-0.7, 0.7-1.0, 1.0-1.5, 1.5-2.0, and 2.0-3.0, with the mean of each bin plotted. Note that there is a

systematic relationship between particle size distribution and AOD, with fine particle median effective radius ($r_c$) increasing with increasing $AOD_{0.675}$.

Figure 4 shows the spectral dependence of the real and imaginary parts of the refractive index for all inversions and sorted as a function of $AOD_{0.675}$. Not all AERONET inversions with size distribution also have refractive index. There are overall fewer retrievals of refractive index and therefore these are grouped into only three bins, with the mean of each bin plotted.

Also, only AERONET refractive index values, with corresponding $AOD_{0.675}$ larger than 0.4 were used in this study to ensure that aerosol signal is significant to retrieve these parameters. This is actually more conservative than the AERONET team recommendations of using inversion products with AOD at 0.44 μm > 0.4 (Holben et al., 2006). Figure 4 shows, unlike size distribution, there is no systematic relationship between refractive index and AOD in this data set. The variability in each AOD bin exceeds the differences between the bins. Thus, we use single mean values for the real and imaginary parts of the

refractive index in our regional aerosol model. Particularly, we calculated averaged refractive index and interpolated to 0.55 μm. The real part of the refractive index is interpolated linearly, while the imaginary part is interpolated using logarithms from 0.44 μm and 0.675 μm (Lee et al., 2017). The lack of AOD dependency in refractive index is possibly due to the limited sample size of this data set that is not representative of the full range of conditions experienced during the season. There are very few AERONET inversion products for $AOD_{0.675}$ > 2.0 during the burning season, and yet from AERONET

direct sun observations of AOD and satellite retrievals we know that the $AOD_{0.675}$ > 2.0 is common. One significant source of uncertainty in the research algorithm being developed here is the extrapolation of these constant refractive indices beyond the range of their formulation data set to represent smoke optical properties for AOD's > 2.0.

Table 1 shows the comparison between the fine mode of the operational model that is used over the Indonesia region and the newly generated smoke model. The natural logarithm of the standard deviation of the radius (σ) and the volume of particles

30  per cross section of the atmospheric column ($V_o$) remain unchanged. However, Indonesian smoke particles are larger and increase more rapidly with AOD than the operational model. The differences in the imaginary part of the refractive index show that Indonesian smoke is substantially less absorbing (whiter) than the generic moderately absorbing model currently employed by the algorithm, especially for very thick smoke plumes. The generic (operational) aerosol model shows increased absorption with increasing AOD, which may represent "brown" smoke better rather than "white" smoke.





However, we note that a widely-used AERONET-derived smoke model from data taken in South America also shows no AOD dependence on its absorption properties (Dubovik et al., 2002). These differences, especially due to the differences in absorption can introduce a retrieval bias in AOD on the order of one. We use this newly generated regional smoke model to generate a research AOD product over Indonesia region during the wildfire season.

Using the regional smoke model and the algorithm with modified masking, we re-produce the AOD for the case study of 22 September 2015, shown in Fig. 5a. This product is referred to as the "research AOD". Compared with Fig. 2b, the research AOD has greater data coverage (availability shown in green in Fig 5b), especially over the regions where optically thick smoke plumes exist. At the center of the plume, the research $AOD_{0.55}$ can be as higher than 5, but is constrained to be 5 because of the lack of sensitivity of the algorithm to very high AOD. Areas with no AOD retrievals within the plume are
identified as clouds. By using the new aerosol model, the retrieval values are altered as well. When the DT $AOD_{0.55}$ is less than 1.0, the two products report very similar retrievals with the differences of less than 0.1 as shown in Fig 5b. When the DT $AOD_{0.55}$ is greater than 1.5, the research algorithm produces smaller $AOD_{0.55}$ values. This is due to the use of a new aerosol model with less absorption.

  This modified research algorithm is tuned to retrieve over optically thick smoke plumes and performs best when these
targeted features exist within the scene. Thus, a pre-selection scheme of MODIS granules is needed to ensure the research algorithm runs on an appropriate granule. To achieve this goal, two parameters are considered: OMI AI for confirming the existence of absorbing aerosols and high AOD values ($AOD_{0.55} > 2.5$) from the operational DT product to ensure that heavy smoke aerosol plumes exist within the scene. Still, those parameters need to be used with caution. A thin layer of absorbing aerosol above clouds can trigger very high AI values especially for regions with optically thick clouds (Meyer et al., 2013;
Yu et al., 2012; Alfaro-Contreras et al., 2014; Torres et al., 2012). Also, erroneously high MODIS AOD can be found over cloud edges due to inaccurate cloud screening or cloud 3-D effects (Zhang et al., 2006; Shi et al., 2010). Utilization of the two parameters together provides better detection of the ideal granules for the study. In order to minimize "fake high aerosol loading" associated with cloud artifacts, we require AI values to be greater than 2.5 and at least five pixels of the operational MODIS DT $AOD_{0.55}$ to be greater than 2.5. All granules are hand-checked from August to October 2015. There are 80
granules that satisfy our selection criteria and contain optically thick smoke plumes that are not available in the operational DT products.

**5 Validation of the research AOD for Indonesian smoke in 2015**

The research algorithm is applied to 80 selected granules. The retrieved AODs from the research algorithm are evaluated against AERONET direct sun AODs and are inter-compared with AODs from the operational DT product. The comparison
is based on spatiotemporal collocations of MODIS retrievals within 0.3° Lat/Lon of the AERONET site location and AERONET observations within 30 minutes of the satellite overpass times. Figure 6 shows the scatter plot of MODIS versus AERONET AODs for (1) the operational DT product, (2) an intermediary retrieval that uses the same masking as the





operational algorithm but implements the new heavy smoke aerosol model, and finally, (3) the research version of the MODIS AOD using the new aerosol model and the modified cloud/NDVI masks (referred as the research algorithm hereafter) along with the error statistics and error envelopes (±0.05±15%AOD) from the operational DT product. As shown in Fig. 6, the distribution of MODIS-derived AOD products are generally correlated with AERONET AOD, with the DT

AOD exhibiting much larger scatter at high AODs. The mean bias in Fig. 6a show that changing the aerosol model reduced the value of retrieved AOD, especially when high AOD exists. This is because the newly generated regional smoke model assumes smoke aerosols as less absorbing than does the generic model used in the operational DT retrievals. That is also the reason for the extra points retrieved when using the new aerosol model: Some retrievals (10 pixels) are greater than 5.0 when using the generic aerosol model and are not reported by the operational algorithm. Applying the new, less-absorbing smoke

model brings those retrievals down into the reportable range. In addition to bringing back previously unreported retrievals, the retrievals from the new aerosol model (Fig. 6a red) have lowered the root mean square error (RMSE) and show higher correlation with AERONET data. Meanwhile the full research algorithm, which uses the new aerosol model and less restrictive masking (Fig. 6b), nearly doubled the number of high AOD retrievals for $AOD_{\text{655}} > 1$, yet yields retrievals with RMSE that is much less than is reported for the operational DT products.

We analyze the satellite-AERONET bias of the DT and research AOD as a function of AERONET AOD, and show the results in Figure 7. In Fig. 7 C6 AOD is binned every 5 pixels with the last bin has 7 pixels and research AOD is binned every 5 pixels with the last bin has 6 pixels. When AERONET AOD is less than 1.5, there is a relatively small positive bias between both MODIS products and the AERONET AOD. When AERONET AOD is greater than 1.5, the bias in the DT AOD grows to around 1.0 while the bias in the research AOD is only roughly half of that. The research product maintains a

mean bias against AERONET of ΔAOD ~ 0.5 or less across the entire range of AERONET AODs, and shows very good agreement (ΔAOD < 0.25) at the very highest AODs ($AOD_{\text{655}} > 3$). We note that the standard deviation of the bias can be large even when the mean bias is low. The regional aerosol model that we used represents non-absorbing white smoke emitted by intense peat burning, which may be the dominant source of the heavy smoke here, but not the only source. When the smoke is produced from open flames or other processes, the optical properties of the regional model will not capture

these differences and biases are introduced. For example, the mean negative bias in the C6 AOD at AERONET $AOD_{\text{655}} > 3$ is partially due to the generic aerosol model used in the operational algorithm that is much more absorbing than the heavy smoke generated in this event. We have used 163 AERONET inversions, independent of the MODIS overpass, to form the research aerosol model, and then validated the resulting research product using AERONET direct sun observations of AOD collocated with MODIS retrievals. Figure 7 shows that for the most part this aerosol model works for the highest loading

type of smoke, but given a larger formulation data base with more AERONET inversions, an aerosol model might be developed that better captures the variability of smoke optical properties during a heavy burning season.

The new research algorithm increases data coverage temporally and spatially. Increase in temporal data coverage in the research product is expected and observed for all five AERONET stations, because most of the sites are influenced by





optically thick smoke around mid to late September. Palangkaraya and Pontianak AERONET sites are located in the central and west parts of Kalimantan, where the most severe burning occurs. Thus, the AOD time series over these two sites show the most significant differences in data coverage between DT AOD and the research AOD. Figure 8 shows the time series of pixel level AERONET observations (in grey), the MODIS DT (in blue) and the research product (in red) over Palangkaraya

and Pontianak sites. MODIS data that are collocated with AERONET observations both spatially and temporally are shown by dots, while crosses show same-day spatial collocations that are not restricted to ±30 minutes of overpass. Unlike Fig. 6, here we plot every individual MODIS retrieval within the 0.3 degree radius circle rather than averaging all the retrievals in the circle and only plotting the mean. Likewise, we plot every AERONET observation, regardless of whether there is a collocation with MODIS over pass. For this exercise only because we note the large sample of AERONET AOD greater

than 5, we also plotted research data that are larger than 5 using open circle and plus sign, respectively. Also note that sunphotometry reaches its limit when AOD equals to 7 multiplied by airmass (7*m). Thus, the gaps in the AERONET $AOD_{0.55}$ time series at Palangkaraya and Pontianak could be because the AOD exceeded this value at this wavelength. Comparison at a longer wavelength, such as 0.675 $\mu$m might have yielded a larger sample because of the small particle size of smoke and corresponding strong spectral dependence would produce AODs less than the AOD limit. However, using

longer AERONET wavelengths would have required spectral extrapolation of the AOD to 0.55 $\mu$m in order to compare with the MODIS product, introducing additional uncertainty.

The time series begins August 28[th] (Julian Day 240) at the onset of severe biomass burning, and proceeds to October 28[th] (Julian Day 300), the conclusion of the heavy burning. MODIS data that are collocated both spatially and temporally with AERONET show instantaneous agreement with AERONET in Fig. 8. As MODIS and AERONET observation times begin

to stray outside the one-hour collocation window (crosses), the MODIS retrievals do not always agree as well with AERONET observations. Over all, we see the MODIS products matching AERONET well, both in terms of day-to-day means and also in terms of spatiotemporal variability. The spread of MODIS points, caused by spatial variability, agrees well with the spread of AERONET points, caused by temporal variability. This agreement supports the use of spatiotemporal statistics in the scatter plots of Fig. 6. The research product provides much more data especially when

AERONET observed AOD is greater than 2.0 and captures the instantaneous high AOD that are observed by AERONET. Over the Palangkaraya site, where the operational product misses most of the burning event, the research product is able to retrieve on many days over this period. A similar pattern can be found over the Pontianak site where the research product provides better data coverage of events with its AOD retrievals following the pattern of the AERONET AOD time series well. We also see several situations where the operational DT values are too high, as compared with AERONET, but the

research algorithm values are less so.

For the limited pairs of collocation data that we have, the research algorithm is producing values of AOD that generally agree with collocated AERONET values, on average. For $AOD_{0.55} \leq 1.0$ the error range is very similar to that from the operational DT algorithm. For all available AOD during this period and domain 48% of the operational DT data points fall within the error bounds defined for the global DT over land algorithm (±0.05±15%AOD; Levy et al., 2013). In comparisons,





66% of the AODs retrieved by the research algorithm fall within this error envelope. If we relax this error bound to 0.05+17%AOD and -0.05-15%AOD, then 70% of the research AOD fall within this range. At the same time, the new research algorithm has doubled the number of retrievals with $AOD_{dss} > 1$.

### 6 Characterization of the Indonesian 2015 burning season

The new research algorithm provides better characterization of the Indonesian fire season because it offers more frequent sampling of the heavy smoke events and better accuracy. Because the research product retrieves high AOD more often than the operational product, the MODIS-derived regional AOD climatology will change. Figure 9 shows the histogram of MODIS AOD over the Indonesian region from August to October 2015 on a logarithmic scale. Here we do not constrain the upper limit of the retrieved AOD. The red is the research AOD and the blue is the DT AOD. When AOD is small the AOD

distributions of the research product and the operational product are very similar. Note that in the -0.1 to 0.0 bin the research AOD (red) matches the DT AOD (blue) and that is why no red bar can be seen. However, the research product has much more data available than the operational product when AOD is greater than 2.0. The number of AOD retrievals in the 4.0 to 5.0 bin almost doubles for the research algorithm, as compared with the operational product, and there are many retrievals of AOD greater than 5.0 in the research product, but none with the operational product. Note that due to removing the upper

bound of $AOD_{dss} = 5.0$ we have allowed the research algorithm to extrapolate beyond the limits of the LUT (Section 2.1), and the research AOD can reach very high values. To maintain the integrity of the AOD histogram, we included values greater than 5 in Fig. 9 and showed them using shaded white lines. However, we do not know the variations or uncertainties of these extremely high AOD retrievals and thus recommended using them with caution. Again, during our validation and analyses, we capped AOD at five.

Monthly mean domain-averaged AOD statistics are shown in Table 2 for both MODIS aerosol retrievals over land. The domain is defined as -10° to 10° N and 95° to 125° E. During August when the severe burning has not yet started, the two AOD products provide similar statistics of AOD. Then during September and October, once the burning has become severe, we see higher monthly mean AOD values over land with the research algorithm than with the operational DT algorithm. The difference in the over land domain averaged AOD is about 0.2 for both months.

Figure 10a and b show the spatial distribution of averaged AOD from the research product and the operational product at 0.5° resolution over the study domain from August to October 2015. The research product shows much more intense smoke in the burning regions of Borneo and the island of Sumatra, than does the operational DT algorithm. However, there is almost no change of retrieved AOD over regions where intense burning did not happen, such as North Borneo and North Sumatra. The differences between Fig. 10a and b are shown in Fig. 10c. Grid boxes with AOD differences greater than 1.0

are found over most of the areas where severe burning occurs. Such large differences are found in 8% of the total land grid boxes in September and 9% in October. At the center of the burning, differences in AOD can be above 3.0. Even over



regions that are mostly influenced by transported smoke, such as Singapore, the research product shows AOD about 0.3 higher than what operational product reports.

**7 Summary and Conclusions**

The MODIS DT aerosol algorithm, developed for "normal" global conditions, exhibits a problem in characterizing the
aerosol during an anomalously severe wildfire season in Indonesia. The DT algorithm misses the heaviest smoke scenes and returns inaccurate values of AOD when it does make a retrieval in heavy smoke. We found that problems of missing or inaccurate reporting of AOD over this event not only existed in the MODIS DT aerosol algorithm but also in other aerosol satellite products as well.

To "save" the optically thick smoke data that the traditional DT product misses, we tune the operational MODIS DT aerosol
algorithm pixel selection routines and develop a regional aerosol model from local AERONET inversion products. One important change is the cloud mask. Based on the particle size differences in smoke and clouds, the MODIS cloud optical properties algorithm will fail when attempting to retrieve cloud microphysical properties from heavy aerosol. So, even if a pre-retrieval screening process cannot separate smoke from clouds, a post-retrieval screening process, based on the failure of the cloud algorithm will make the distinction. We make use of the cloud product failure metrics to bring back heavy smoke
pixels at the center of the smoke plumes. A second important change is the regional smoke model generated from AERONET inversion products that better represents aerosol properties in this region. The generated smoke model is a function of AOD for particle size distribution, but not for absorption properties. The AERONET inversion products are analyzed from a limited data set that includes few retrievals for $AOD_{0.55} > 2.0$. Thus, extrapolation of aerosol particle properties to the highest AOD situations introduces uncertainty, especially for the absorption properties which exhibit no
AOD dependencies for $AOD_{0.55} < 2.0$, but may for $AOD_{0.55} > 2.0$. We do note that the standard deviation of the bias between AERONET and the research AOD at high AOD can be large even when the mean bias is low, possibly suggesting multiple types of smoke optical properties in the data set. We look forward to future datasets from AERONET which may provide additional constraints on absorption and other optical properties during high aerosol loadings.

This research algorithm is designed to retrieve very optically thick smoke and is applied only on MODIS granules that
contain the targeted feature. Thus, a pre-selection procedure is used to select suitable granules for the study. The pre-selection criteria are based on OMI AI, which indicates the existence of absorbing aerosols, and the operational DT AOD to filter out false high AOD due to cloud adjacent effects and situations with aerosol above clouds. The research algorithm is applied to all 80 selected granules over the study region within the study time period. Validation of the research product is done using AERONET version 3 level 2 AOD. The comparisons show that the research product captures more AOD when
$AOD_{0.55} > 1.0$ than does the operational DT algorithm. The research AOD agrees better with AERONET values, resulting in smaller RMSEs and higher correlation statistics. Most of the improvement is found for AOD > 1. On average, 66.3% of the collocated research AOD agree with the current error bounds determined from global analysis of the DT retrievals, which is




much better than 48.4% from the operational AOD in this region. Thus, the research AOD over extreme high smoke loading conditions has nearly the same accuracy as DT product validated over the entire globe. If we relax the error envelopes' upper bound from 0.05+15%AOD to 0.05+17%AOD then 70% of research AOD fall within the bounds. This improvement occurs as the new algorithm doubles the number of retrievals with AOD > 1.

The ability to now retrieve these optically thick smoke plumes alters our understanding of the aerosol system in this region. Statistical analyses illustrate the severe intensity of the monthly and seasonal mean AOD in the specific areas of the heavy smoke, and also show the temporal frequency that was missed with the operational DT algorithm. Using the new algorithm, the domain-averaged over-land $AOD_{0.55}$ increases by 0.22 in September and October of 2015, but over regions where severe burning occurs, the new algorithm increases $AOD_{0.55}$ *by as much as 3.0* for each 0.5° grid box, over the previous operational
algorithm values.

This amount of missing AOD can skew the perception of the severity of the event by researchers and decision-makers who rely on the global DT aerosol for characterization of aerosol systems. The missing AOD can also significantly affect estimates of observationally-based regional aerosol forcing, and improperly influence assimilation systems that rely on the MODIS DT product. Estimating the perturbation of extra AOD on regional radiative balance although beyond the scope of
this work, is an obvious task for future study. The ability to bring back the missing retrievals and assure their accuracy with a regionally appropriate aerosol model is an important step in the development of the DT algorithm. However, there are still many steps before this promising research can become an operational application. First, we do not know whether the changes made and validated for the 2015 season will hold in subsequent seasons. Second, we do not know whether the tuning of the algorithm for the Indonesian region will hold for other situations of heavy smoke from wildfires. The
Indonesian smoke proved to be relatively non-absorbing, which might be similar to smoke from peat burnings in other places such as those from Alaskan fires in summer of 2004 and 2005 (Eck et al., 2009), but may be inappropriate for more absorbing smoke in other places and situations. Third, there is also a philosophical question of how fragmented should a global aerosol retrieval become? If there are too many special situations, the product loses its global uniformity. However, the DT algorithm team has already begun the move towards specially tuned situations as they have implemented a special
handling of urban surfaces (Gupta et al., 2016). A specific set of assumptions triggered by heavy smoke is a likely candidate for the next DT specialty retrieval, but first we must prove its global applicability.

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

**Table 1: Optical properties of the aerosol model used by the operational DT algorithm over the Indonesian region and the regional smoke model generated in this study using AERONET inversion products August to October 2015. Diagonal line means no change is made between the two aerosol models.**

| Model | $r_v$, µm | σ | $V_0$, µm³/µm² | Real part of Refractive Index | Imaginary part of Refractive Index |
|---|---|---|---|---|---|
| Moderate absorbing | $0.020\tau + 0.145$ | $0.1365\tau + 0.374$ | $0.1642\tau^{0.775}$ | 1.43 | $-0.002\tau - 0.008$ |
| Regional smoke | $0.040\tau + 0.160$ | | | 1.47 | -0.0038 |

**Table 2 Domain-averaged (-10° to 10° N and 95° to 125° E) monthly mean MODIS-derived AOD at 0.55 µm over land for the operational (DT) and research (Res) algorithms.**

| Months | DT Land | Res Land |
|---|---|---|
| August | 0.41 | 0.41 |
| September | 0.99 | 1.22 |
| October | 1.26 | 1.51 |





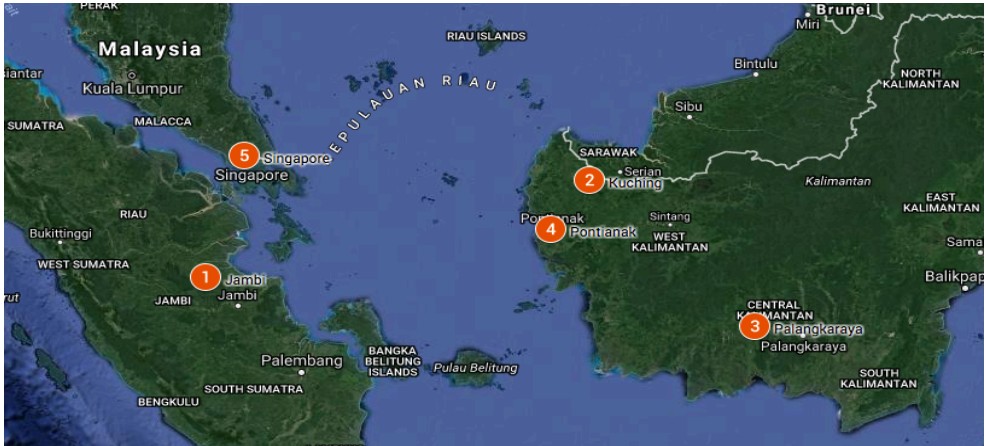

**Figure 1: Locations of the five AERONET sites that are used in this study from google map. 1 Jambi (-1° N, 103° E), 2 Kuching (1° N, 110° E), 3 Palangkaraya (-2° N, 113° E), 4 Pontianak (0° N, 109° E), and 5 Singapore (1° N, 103° E).**





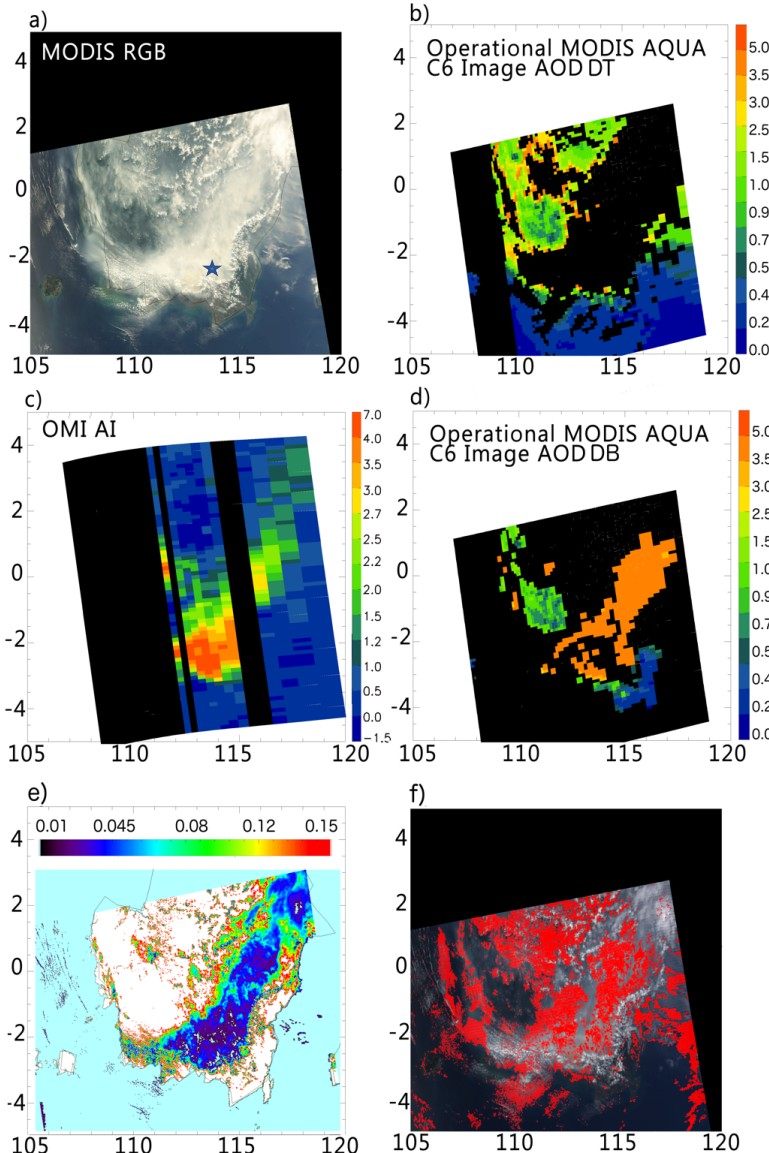

**Figure 2: A case study of a fire in Kalimantan on the island of Borneo in Indonesia on 22 September, 2015. a) RGB image, b) MODIS DT operational image AOD, c) OMI AI, d) MODIS DB all available AOD, e) NDVI value, and f) cloud product failure metrics taken from the MODIS cloud optical properties product. In e) NDVI values smaller than 0.01 are shown in light aqua and values greater than 0.15 are shown in white. In f) the red denotes pixels where the cloud product has failed. These are overlain on top of the MODIS true color image. The blue star represent the AERONET site Palangkaraya.**

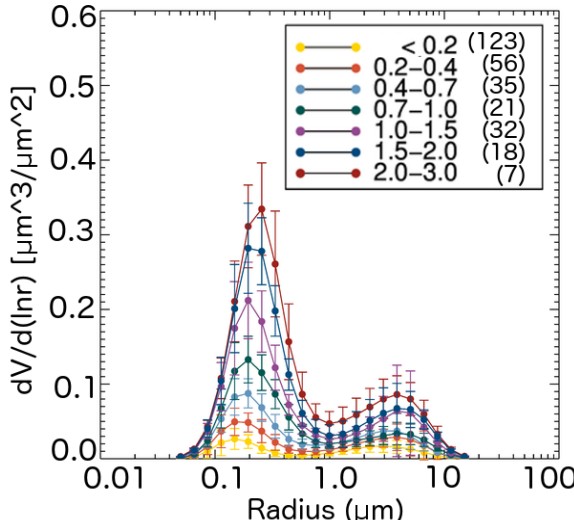

**Figure 3 Size distribution as a function of AERONET AOD at 0.675 μm, generated from the AERONET inversion products at the five sites in Indonesia during August to October 2015. There are 163 total retrievals used in this plot separated into bins of AOD. The number of retrievals within each AOD bin is shown in parenthesis in the label. The error bars represent the standard deviation within each size bin.**

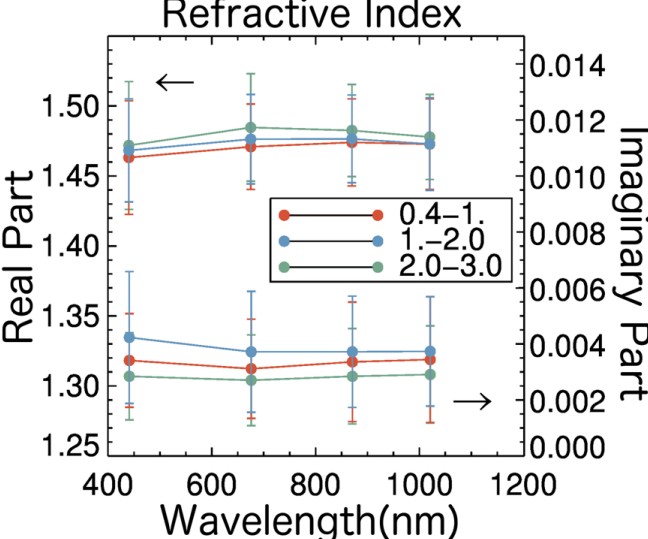

**Figure 4 The real and imaginary parts of refractive index as a function of AOD at 0.675 μm calculated from the AERONET inversion product over the five sites in Indonesia during August to October 2015. The error bars represent the standard deviation within each wavelength. A total of 113 data points are used to generate this plot, 56 retrievals in AOD 0.4-1.0, 50 retrievals in AOD 1.0-2.0, and 7 retrievals in AOD 2.0-3.0.**



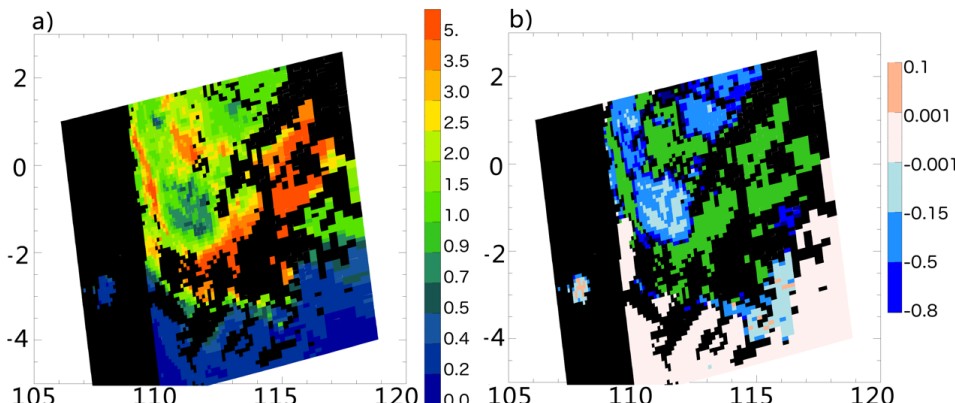

**Figure 5: a)** Research AOD at 0.55 µm retrieved from the case study of 22 September 2015 using altered thresholds on the NDVI test, cloud mask, upper bound limits of the retrieval and a new regional aerosol model. **b)** The differences between the research AOD (panel a) and the DT AOD (Fig 2b). The increased research AOD data coverage is shown in green.

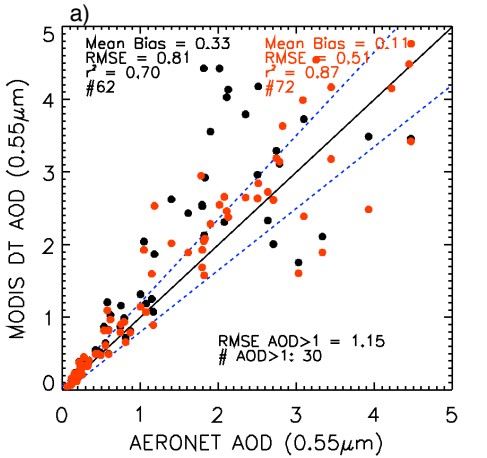

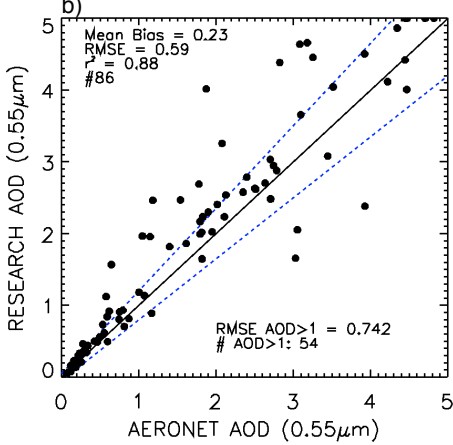

**Figure 6: a)** Comparisons of the MODIS DT AOD at 0.55 µm (black dots) and an intermediate AOD retrieved using the new aerosol model, but same masking as the MODIS DT algorithm (red dots). **b)** AOD at 0.55 µm retrieved by the full research algorithm, all plotted against collocated AERONET observations at five AERONET sites August to October 2015. Also shown are RMSE, correlation coefficient ($r^2$) and number of collocations (#) for the entire range of AODs (upper left and right) and also for a subset of the collocations when AERONET is greater than 1 (lower right). The blue dashed lines are the error envelopes of ±0.05±15%AOD.





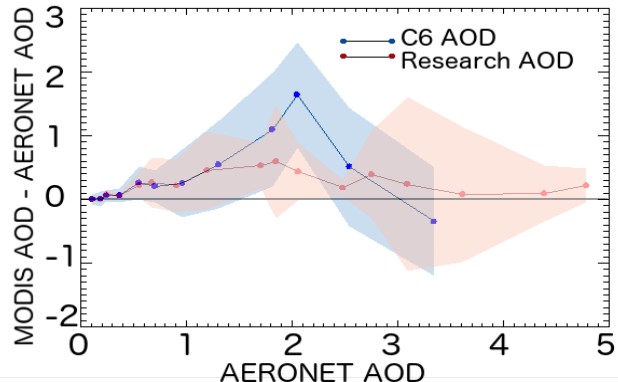

**Figure 7: Bias between MODIS and AERONET over land AOD at 0.55 µm as function of AERONET AOD at 0.55 µm. Blue represents the operational DT AOD and red represents the research AOD. The dots are the mean bias within each AERONET AOD bin and the shaded area represents the standard deviation of the bias.**

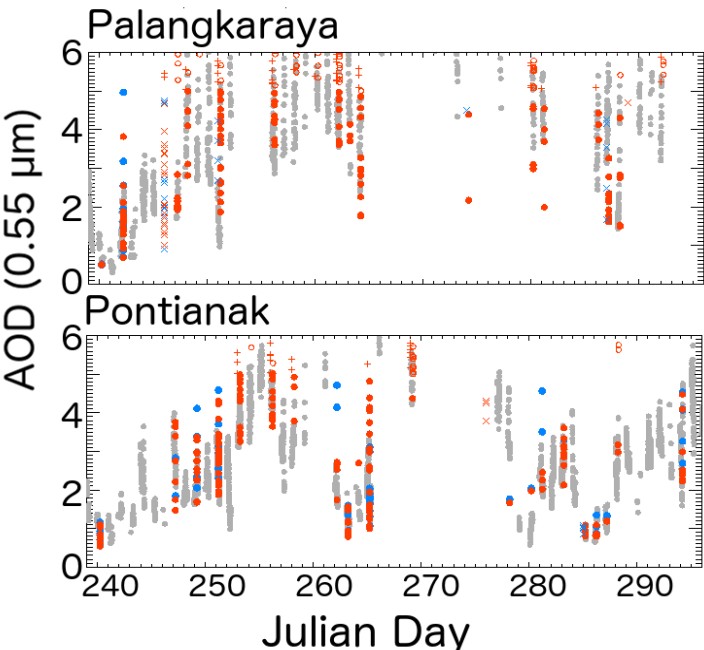

**Figure 8: Time series of all AERONET observations of AOD at 0.55 µm (in grey) as a function of Julian Day in 2015, and the corresponding MODIS DT (in blue) and research (in red) AOD over the Palangkaraya and Pontianak AERONET sites. MODIS data that are temporally and spatially collocated with AERONET are shown by dots, data with values greater than 5 are shown by**





open circle. Same-day spatial collocations that are not restricted to ±30 minutes of overpass are shown by crosses and data with values greater than 5 are shown by plus sign. All individual MODIS retrievals within 0.3° of the AERONET site are included on the plot without averaging.

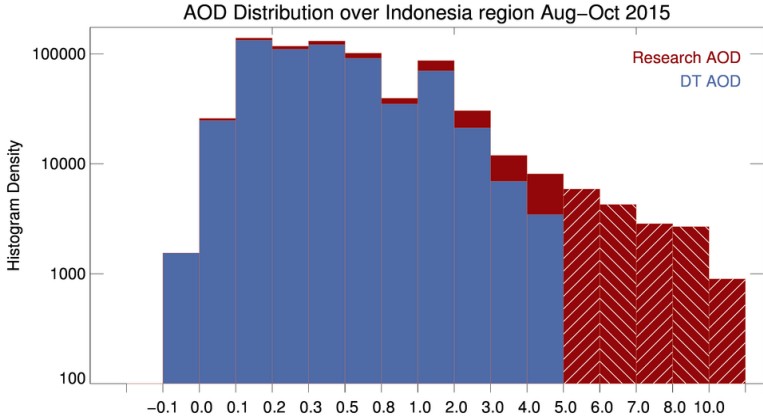

5   Figure 9: The histogram of MODIS AOD over the Indonesia region from August to October 2015 in a logarithmic scale. The red is the research AOD, the blue is the operational AOD. Data that are greater than 5 from the research AOD are shaded using white lines.

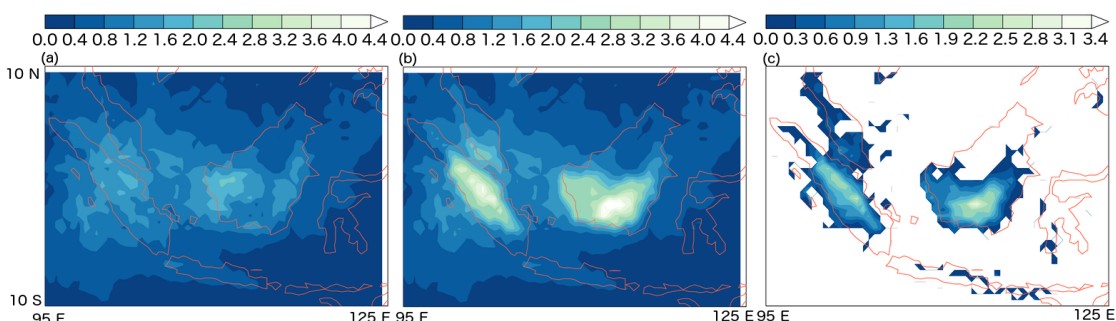

10   Figure 10: Spatial distribution of averaged AOD from the a) the operational product, b) the research product and c) the differences of b) minus a) at 0.5° resolution over the study domain from August to October 2015.