# Peer review of "Characterizing the 2015 Indonesia Fire Event Using Modified MODIS Aerosol Retrievals"

_Atmospheric Chemistry and Physics, 2018_

## Referee Comment (RC1) · Anonymous Referee #1 · 18 Jul 2018

Review of "Characterizing the 2015 Indonesia Fire Event Using Modified MODIS Aerosol Retrievals" by Shi et al. Based on the Indonesian fire and smoke event in 2015, the paper identified the problems in the MODIS DT aerosol algorithm, and proposed solutions to further improve the global DT algorithm. The paper is well written with sufficient technical information and the improvements to the algorithm is evidently clear. I have conducted similar investigation to the VIIRS aerosol products and algorithm, and found very similar results with the VIIRS aerosol algorithm that is based on MODIS heritage DT algorithm. I think the paper outlined a very important issue with the current operational satellite aerosol algorithm, and the proposed improvements are really important for the satellite aerosol remote sensing community. I found the paper suitable for ACP and I recommend its publication in current format. I only have a few

minor comments and suggestions for consideration: 1. Page 3 Line 25, Equation 1: Better have bracket to avoid any potential confusions 2. Page 4 Section 2.2: DB algorithm is introduced here. It will be nice to briefly mention how this study will make use of the DB algorithm for comparison. For example, similar to the last paragraph in Section 2.3. 3. Page 20 Table 1: highlight less absorbing in the 'Regional Smoke' model name, maybe 'Regional Less Absorbing Smoke Model'? 4. Figure 2 is very interesting. DB product also missed quite some retrievals in comparison to DT product. Is it attributable to their different cloud screening? From the RGB image, it looks like to me that DT is sort of underscreening clouds but DB is overscreening. Very intriguing. It seems AI which uses UV channels has a better coverage when aerosols are above clouds. I wonder what OMI AOD will look like for the same scene. 5. Figure 2(e) and (f): it will be nice to put 'NDVI' title on the figure same as you did for (a) to (d) 6. Figure 2: do we have similar Figure 2(g) that shows where in-land water test failed? 7. Figure 8 needs legends in addition to your caption descriptions 8. Figure 9: Are there any bin that blue bars are actually taller than red bars? If yes, you may consider using transparency 9. Figure 10: Why this Figure cannot use the same colorbar as the conventional rainbow colorbar in Figure 2 (c)? 10. It will be nice to show how many missing retrievals are due to cloud overscreening and how many are due to in-land water overscreening, in a Table or in a chart

---

## Referee Comment (RC2) · Anonymous Referee #2 · 17 Sep 2018

Dear Authors,

Thank you for a well-written manuscript on your interesting and carefully performed study. The limitations of operational satellite retrievals for extreme events are well-known but had never been exactically quantified.

I have a few questions seeking clarity, and several minor corrections, but I think with minor revisions this study can be published. Thank you for your hard work.

Comments below are organized roughly by significance:

Q: Would you say that your research results represent a lower bound on the low bias associated with the operational DT retrieval?

[Figure]

Q: The MODIS Dark Target retrieval makes some assumptions about the impact of aerosol scattering and absorption at longer wavelengths. For AE=1.8, AOD550 of 5.0 corresponds to an AOD2.1um of 0.45, which is quite substantial. I think you should discuss the ramifications of high AOD values on the assumptions of the MODIS DT retrieval, especially the reflectance ratios of longer wavelengths.

P5L29 and many other places. I don't like the term "failure metrics," because nothing is actually being measured. My preference would be to refer to this as the "cloud optical properties product MYD06 Collection 6 diagnostic quality flags," which is long but eliminated ambiguity. "Failure metrics" should be replaced with "diagnostic flags" or "quality flags" throughout the manuscript.

P11L26: "an aerosol model might.. better capture the variability of smoke optical properties" What kinds of improvement would you expect to see with more data? What kinds of conditions are undersampled with the existing dataset? Do you have suspicions about how your current results may be biased?

P14L23: "multiple types of smoke optical properties" Are you suggesting there may be multiple modes of smoke particle optical properties? Or are you only saying that the smoke particle optical properties are highly variable?

Figure 9. I recommend modifying the legend (and others) to read "C6 DT AOD" to assist future readers.

Page 2 line 26: "This new regional aerosol climatology" Is this referring to the updated empirical optical properties derived from AERONET? This should be clarified.

P4L13 "information about the aerosol optical properties"

P6L25 "Holben 2006 recommends a threshold of AOD>0.4 at 0.44 um for quality assurance of the AERONET inversion products; we followed the procedures of Holben, but used a higher more strict AOD threshold of AOD>0.4 at 0.675 um."

P11L15: "with 7 pixels in the last bin"

P15L4: This last sentence can be phrased better, I think. If I understand, you are trying to say that the research retrieval has more than double the number of AOD>1, and with those additional retrievals included, the bulk error statistics still show a large improvement.

---

## Author Response (AR1)

Reply to reviewer #1:

*Review of "Characterizing the 2015 Indonesia Fire Event Using Modified MODIS Aerosol Retrievals" by Shi et al. Based on the Indonesian fire and smoke event in 2015, the paper identified the problems in the MODIS DT aerosol algorithm, and proposed solutions to further improve the global DT algorithm. The paper is well written with sufficient technical information and the improvements to the algorithm is evidently clear. I have conducted similar investigation to the VIIRS aerosol products and algorithm, and found very similar results with the VIIRS aerosol algorithm that is based on MODIS heritage DT algorithm. I think the paper outlined a very important issue with the current operational satellite aerosol algorithm, and the proposed improvements are really important for the satellite aerosol remote sensing community. I found the paper suitable for ACP and I recommend its publication in current format. I only have a few minor comments and suggestions for consideration:*

We thank the referee for your nice words.

1. *Page 3 Line 25, Equation 1: Better have bracket to avoid any potential confusions*

   Done.

2. *Page 4 Section 2.2: DB algorithm is introduced here. It will be nice to briefly mention how this study will make use of the DB algorithm for comparison. For example, similar to the last paragraph in Section 2.3.*

   Done. We added "In this study we used DB in a case study to illustrate that both aerosol products (DT and DB) from MODIS have problems retrieving a complete image of AOD when optically thick smoke exists."

3. *Page 20 Table 1: highlight less absorbing in the 'Regional Smoke' model name, maybe 'Regional Less Absorbing Smoke Model'?*

   Done. We changed "Regional Smoke" to "Regional Less-Absorbing".

4. *Figure 2 is very interesting. DB product also missed quite some retrievals in comparison to DT product. Is it attributable to their different cloud screening? From the RGB image, it looks like to me that DT is sort of underscreening clouds but DB is overscreening. Very intriguing. It seems AI which uses UV channels has a better coverage when aerosols are above clouds. I wonder what OMI AOD will look like for the same scene.*

   We agree that the differences in DB and DT retrieval coverage are at least partially due to different cloud masking. However, we don't know the DB algorithm well enough to say that the cloud mask is the only source of differences. We checked the cloud filtering in this image and do not believe that DT is underscreening clouds over this region. To answer your question, we plot OMI AOD (right). Note that OMI AI retrieves over clouds but OMI AOD requires cloud free scenes in its 13x24 km footprint.

[Figure]

5. *Figure 2(e) and (f): it will be nice to put 'NDVI' title on the figure same as you did for (a) to (d)*

Done. We put NDVI title on the figure.

6. *Figure 2: do we have similar Figure 2(g) that shows where in-land water test failed?*

   Figure 2e shows where in-land water test failed, which are colors cooler than yellow. In another word, colors in green, blue, purple, black and light aqua are areas that failed the test. We added "In Fig. 2e only regions colored white and red passed the NDVI threshold" in Page 7 line 15.

7. *Figure 8 needs legends in addition to your caption descriptions*

   Done. We added legends.

8. *Figure 9: Are there any bin that blue bars are actually taller than red bars? If yes, you may consider using transparency*

   No, there is no blue bars actually taller than red bars.

9. *Figure 10: Why this Figure cannot use the same colorbar as the conventional rainbow colorbar in Figure 2 (c)?*

   We used this due to author's personal preference.

10. *It will be nice to show how many missing retrievals are due to cloud overscreening and how many are due to in-land water overscreening, in a Table or in a chart.*

    I understand what the referee is looking for. However, most of the retrievals at the center of the plume were screened out due to both cloud masks and in-land water mask. In our case study, 976 missing retrievals are due to cloud screen, 878 missing retrievals are due to inland water mask. The total number of increases is 1004 (the union of two masks), most of which are screened out by both masks.

Reply to reviewer 2:

*Dear Authors,*
*Thank you for a well-written manuscript on your interesting and carefully performed study. The limitations of operational satellite retrievals for extreme events are well known but had never been exactingly quantified. I have a few questions seeking clarity, and several minor corrections, but I think with minor revisions this study can be published. Thank you for your hard work.*
*Comments below are organized roughly by significance:*

We thank the reviewer for your kind words and warm encouragement. We are happy that you think this work is interesting and important.

1. *Q: Would you say that your research results represent a lower bound on the low bias associated with the operational DT retrieval?*

   We don't really understand this question. Our research product reduces the systematic low bias when compared against AERONET, especially at high AOD levels.

2. *Q: The MODIS Dark Target retrieval makes some assumptions about the impact of aerosol scattering and absorption at longer wavelengths. For AE=1.8, $AOD_{550}$ of 5.0 corresponds to an $AOD_{2.1um}$ of 0.45, which is quite substantial. I think you should discuss the ramifications of high AOD values on the assumptions of the MODIS DT retrieval, especially the reflectance ratios of longer wavelengths.*

   In DT algorithm we match the measured top of atmosphere reflectance at 2.1, 0.66, and 0.47 $\mu m$ with modeled reflectance. The surface reflectance at 2.1 $\mu m$ as well as the contribution from aerosol is solved through equations (ATBD equation 42 a,b, and c https://darktarget.gsfc.nasa.gov/atbd/land-algorithm). We do not have to assume that the atmosphere is transparent at 2.1, only that the three wavelengths are related when solving the equations. In other words, the equations account for the influence of the aerosol loading at 2.1 $\mu m$, although AERONET makes measurements of aerosol in visible and NIR wavelengths (which are used to derive TOA aerosol properties), there are no real measurements of aerosol properties at 2.1 mm. We do not yet have a constraint on "how uncertain" this means for deriving the TOA reflectance at 2.1

3. *P5L29 and many other places. I don't like the term "failure metrics," because nothing is actually being measured. My preference would be to refer to this as the "cloud optical properties product MYD06 Collection 6 diagnostic quality flags," which is long but eliminated ambiguity. "Failure metrics" should be replaced with "diagnostic flags" or "quality flags" throughout the manuscript.*

   We changed "failure metrics" to "diagnostic flags".

4. *P11L26: "an aerosol model might ... better capture the variability of smoke optical properties" What kinds of improvement would you expect to see with more data? What kinds of conditions are under-sampled with the existing dataset? Do you have suspicions about how your current results may be biased?*

   The AOD dependency in the aerosol model is limited due to the lack of high AOD data in AERONET inversion data base in this study. We would like to have more AERONET inversion data in high AOD regimes, for example when AOD is greater than 3. With limited information

about aerosol properties when the AOD is high, we don't know if or how the absorption will change in higher AOD regimes. If the higher AOD means proportionally more scattering than absorption than our current extrapolation of particle properties suggest, then our retrieved high AOD will be biased high. If at a certain AOD, the scattering/absorption plateaued, then the current extrapolation is accurate and our retrieval will have less bias. Also, currently we are only able to create a model for the more prevalent white peat-burning smoke over our study domain. With more data, we can separate the non-absorbing white smoke from the brown smoke that was emitted by the flaming, and reduce the bias in our retrieval introduced by using the white-smoke model for situations that require a greater component of absorption. We added couple sentences as part of the uncertainty discussion in the end of chapter 5.

5. *P14L23: "multiple types of smoke optical properties" Are you suggesting there may be multiple modes of smoke particle optical properties? Or are you only saying that the smoke particle optical properties are highly variable?*

   We changed this sentence to "multiple types of smoke with different absorbing properties"

6. *Figure 9. I recommend modifying the legend (and others) to read "C6 DT AOD" to assist future readers.*

   Done.

7. *Page 2 line 26: "This new regional aerosol climatology" Is this referring to the updated empirical optical properties derived from AERONET? This should be clarified.*

   We changed this sentence to "Using the newly developed research product, we investigated how our regional climatology was modified."

8. *P4L13 "information about the aerosol optical properties"*

   Done

9. *P6L25 "Holben 2006 recommends a threshold of AOD>0.4 at 0.44 um for quality assurance of the AERONET inversion products; we followed the procedures of Holben, but used a higher more strict AOD threshold of AOD>0.4 at 0.675 um."*

   Done

10. *P11L15: "with 7 pixels in the last bin"*

    Done

11. *P15L4: This last sentence can be phrased better, I think. If I understand, you are trying to say that the research retrieval has more than double the number of AOD>1, and with those additional retrievals included, the bulk error statistics still show a large improvement.*

    We changed this sentence to "the research retrieval has more than double the number of AOD>1, and with those additional retrievals included, the bulk error statistics still show a large improvement."

[revised manuscript text omitted]

The research algorithm increased the number of retrievals and reduced the bias against AERONET measurements. However, the method retains some sources of uncertainty, which contribute to errors in the retrieval. One is that the AOD at 2.1 $\mu m$ can be high. In DT algorithm we match the measured top of atmosphere (TOA) reflectance at 2.1, 0.66, and 0.47 $\mu m$ with modeled reflectance. The surface reflectance and contribution of aerosols at 2.1 $\mu m$ are counted for. This method is most accurate when the influence of the aerosols at 2.1 $\mu m$ is negligible. Hence, having a high AOD at 2.1 $\mu m$ will still influence the retrieval accuracy, because assumptions made about aerosol optical properties influence the partition between surface and atmosphere contributions to the TOA signal. Although AERONET makes measurements of aerosol in visible and near-IR wavelengths, which are used to derive the modelled TOA reflectance, it does not provide aerosol properties at 2.1 $\mu m$ and there are few/no alternative sources. Thus, we do not yet have a constrain on the uncertainty of derived TOA reflectance at 2.1 $\mu m$, and a high aerosol loading at this wavelength will enlarge the uncertainties.

Another uncertainty source is in the aerosol model parameterization when AOD > 3. Our study shows it is important to have an AOD-dependent aerosol optical model to reduce the bias when retrieving AOD under very high aerosol loading. Having more measurements of aerosol optical properties at AOD > 3 can further reduce our retrieved uncertainties. However, a perfect AOD dependent peat burning smoke aerosol model is still not adequate to represent every smoke plume. Smoke properties vary largely due to the type of burning, and although peat burning (which tends to appear white) dominated the 2015 Indonesia fire event, there were still brown smoke plumes seen occasionally, which were caused by open flaming. A fixed regional aerosol model introduces bias when different type of burning occurs. Thus, an instantaneous retrieval of aerosol absorption is the key to get more accurate retrievals at very high aerosol loading, and further research in this direction is needed.

[revised manuscript text omitted]